# Differential Effects of Human Tau Isoforms to Neuronal Dysfunction and Toxicity in the Drosophila CNS

**DOI:** 10.3390/ijms232112985

**Published:** 2022-10-26

**Authors:** Ergina Vourkou, Vassilis Paspaliaris, Anna Bourouliti, Maria-Christina Zerva, Engie Prifti, Katerina Papanikolopoulou, Efthimios M. C. Skoulakis

**Affiliations:** 1Institute for Fundamental Biomedical Research, Biomedical Sciences Research Centre “Alexander Fleming”, 16672 Vari, Greece; 2Department of Neurology, School of Medicine, National and Kapodistrian University of Athens, 11527 Athens, Greece; 3Second Department of Neurology, “Attikon” General University Hospital, 12462 Athens, Greece; 4Laboratory of Experimental Physiology, School of Medicine, National and Kapodistrian University of Athens, 11527 Athens, Greece; 5Department of Molecular Biology and Genetics, Democritus University of Thrace, 68100 Alexandroupolis, Greece; 6Athens International Master’s Program in Neurosciences, Department of Biology, National and Kapodistrian University of Athens, 15784 Athens, Greece; 7Department of Biotechnology, Agricultural University of Athens, 11855 Athens, Greece

**Keywords:** tau isoforms, learning and memory, habituation, neuronal dysfunction, toxicity, Drosophila

## Abstract

Accumulation of highly post-translationally modified tau proteins is a hallmark of neurodegenerative disorders known as tauopathies, the most common of which is Alzheimer’s disease. Although six tau isoforms are found in the human brain, the majority of animal and cellular tauopathy models utilize a representative single isoform. However, the six human tau isoforms present overlapping but distinct distributions in the brain and are differentially involved in particular tauopathies. These observations support the notion that tau isoforms possess distinct functional properties important for both physiology and pathology. To address this hypothesis, the six human brain tau isoforms were expressed singly in the Drosophila brain and their effects in an established battery of assays measuring neuronal dysfunction, vulnerability to oxidative stress and life span were systematically assessed comparatively. The results reveal isoform-specific effects clearly not attributed to differences in expression levels but correlated with the number of microtubule-binding repeats and the accumulation of a particular isoform in support of the functional differentiation of these tau isoforms. Delineation of isoform-specific effects is essential to understand the apparent differential involvement of each tau isoform in tauopathies and their contribution to neuronal dysfunction and toxicity.

## 1. Introduction

Tau is an abundant central nervous system (CNS) protein, largely localized on the labile domains of microtubules, promoting their elongation and thus regulation of axonal transport [1]. Despite its initial characterization as an axonal protein, recent evidence revealed that tau is also present, albeit at low levels, in the somatodendritic compartments, sub-synaptic sites, and the nucleus suggesting additional physiological roles to its standard function in cytoskeletal regulation [2].

Alternative splicing of the primary transcript from the single human *MAPT* gene on chromosome 17 gives rise to 6 tau isoforms in the adult CNS and two isoforms in the peripheral nervous system [3,4]. The brain-specific isoforms vary in the number of N-terminal inserts (0N, 1N, or 2N) and C-terminal repeat domains (3R or 4R) due to the inclusion or omission of exons 2, 3 and 10, resulting in protein sizes from 48 kDa (shortest 0N3R) to 67 kDa (longest 2N4R). Because of the extra microtubule-binding repeat, 4R isoforms ostensibly bind more effectively to microtubules and promote their assembly more potently than their 3R counterparts [5]. However, the number of microtubule-binding repeats does not seem to regulate tau axonal sorting [6]. The diversity of tau isoforms is further increased by potentially differential post-translational modifications including phosphorylation, glycosylation, ubiquitination, nitration, deamidation, oxidation and glycation [7].

Tau expression is developmentally regulated, with 0N3R being the sole isoform in the developing human embryonic brain. In contrast, all isoforms are expressed in the adult CNS with almost an equimolar ratio of 3R to 4R species. However, 1N, 0N and 2N isoforms are not equally represented, but rather account for 54%, 37% and 9% of total tau, respectively [4,8]. The subcellular distribution of tau seems to be isoform-specific. Hence, 2N isoforms show a higher propensity for somatodendritic localization [6], while 1N isoforms are abundant in the nucleus [9,10] and 0N isoforms are detectable both in the soma and axon [2]. Furthermore, tissue-specific splicing differences of the *MAPT* transcript appear to account for the lower amount of 0N3R in the cerebellum compared to other brain regions and the elevation of 4R isoforms in the globus pallidus of the basal ganglia [11,12]. These observations suggest neuronal type-specific functions of the isoforms and may also reflect functional differentiation among the isoforms as well.

Although the functional implications of this differential distribution of tau isoforms are still unclear, the 3R/4R isoform balance appears critical for brain function since its disruption is a hallmark of tauopathies [7]. Tauopathies encompass a range of neurodegenerative disorders characterized by the presence of intraneuronal fibrillar inclusions of highly phosphorylated forms of tau. Some familial tauopathies are consequent to *MAPT* mutations resulting in excess exon 10 inclusion by splicing dysregulation and thus elevation of 4R isoforms [13]. Furthermore, altered 3R/4R ratios are observed in the CNS of largely non-familial Alzheimer’s disease (AD) patients [14]. Altered 3R/4R ratios appear to also be reflected in tau aggregates, leading to the categorization of tauopathies on the basis of whether filamentous inclusions are composed largely of 3R, 4R or both types of isoforms [15]. Therefore, it is essential to elucidate any molecular and physiological differences among isoforms towards gaining insights into their apparent differential involvement in distinct tauopathies and their contribution to neuronal dysfunction and toxicity.

Functional differences between 4R and 3R isoforms have been revealed in vitro, focusing mostly on how they alter microtubule dynamics [6,16,17]. In addition, upon isoform imbalance, cultured neurons presented mitochondrial axonal transport defects [18] and impaired axonal transport of amyloid precursor protein [19]. Human tau isoform-specific phenotypes in vivo have been described mostly in Drosophila tauopathy models by assessing neuronal function (including axonal transport, locomotor behavior, synaptic function, olfactory learning and memory) [20,21,22] and cell loss/toxicity during development (photoreceptor neuron and mushroom body neuron loss), or in adult animals (elevated oxidative stress, shortened lifespan) [22,23,24]. For example, the 3R isoform in Drosophila resulted in more severe axonal transport abnormalities, locomotor deficits and a shorter lifespan than 4R [22]. In contrast, the 4R isoform caused more severe neurodegeneration and learning and memory deficits, which were not observed in 3R-expressing animals [22,24].

Even though these differences may reflect *bona fide* distinct physiological functions and pathogenic potential among these two tau species, the remaining isoforms were not examined in these assays. Hence, it has been unclear whether these differences generalize to the remaining 3R and 4R isoforms or are specific to 0N3R and 0N4R [22]. In addition, such differences among isoforms could arise because of differences in the expression of tau-encoding transgenes inserted in different genomic locations (position effects). Herein, we systematically examine the effect of all six CNS-expressed human isoforms in the Drosophila CNS for deficits in tauopathies relevant neuronal function, toxicity and premature lethality assays. This is facilitated by the availability of strains where all human tau isoform-encoding transgenes were landed at the same chromosomal site and their cDNA was codon optimized for expression in Drosophila [25]. Collectively, the results support the notion that the CNS-expressed human tau isoforms are not functionally equivalent with respect to their pathogenic potential, probably reflecting differences in their physiological function as well.

## 2. Results


**Distinct accumulation levels of the transgenic hTau isoforms in the fly CNS.**


To unravel in an unbiased manner the effect of each human tau (hTau) isoform in vivo, we used transgenic Drosophila lines generated by a site-directed integration strategy on chromosome II and reported to ensure comparable expression levels under the double *nSyb*-GAL4 driver [25]. To obtain comparable results with prior work [24,26,27], the transgenes were expressed under the *elav^C155^*-GAL4 (henceforth elavG4) and, to our surprise, they yielded variable tau levels (Figure 1A) in multiple experiments from different crosses. The quantification of three independent blots revealed statistically significant differences in the levels of hTau^0N3R^, hTau^2N3R^ and hTau^2N4R^ compared to hTau^1N3R^, hTau^0N4R^ and hTau^1N4R^ (Appendix A). hTau^0N3R^, hTau^2N3R^ and hTau^2N4R^ levels were not significantly different from each other as were those of hTau^1N3R^, hTau^0N4R^ and hTau^1N4R^. These results are in broad agreement with those of Fernius [25] but offer better resolution, possibly because of the strong pan-neuronal expression throughout development and adulthood under elavGal4. Therefore, hTau levels comprise two abundance groups (Figure 1A), with their levels not correlating with their size, N or R repeats. These unexpected results for single-site inserted transgene-encoded proteins under the same Gal4 driver could reflect mRNA instability, differences in translation efficiency or differential instability of the hTau isoforms in the fly CNS.

To address these possibilities, the steady-state transgenic mRNA levels were quantified (Figure 1B) and were not found to be statistically different (Appendix A). Therefore, the differential hTau isoform accumulation may reflect differences in the translational efficiency of their mRNAs or in the stability of the resultant proteins. Given that the transgenic transcripts were codon optimized for Drosophila, it is unlikely that the differences in hTau isoform accumulation result from the differential translatability of their encoding transgenic RNAs. Therefore, it appears that the hTau isoforms are differentially stable in the Drosophila CNS.


**Differential effects of hTau isoforms in the development of mushroom body neurons.**


The mushroom bodies (MBs) are bilaterally symmetrical, structurally stereotypical neuronal assemblies in the fly CNS [28]. The MBs constitute major insect brain centers for learning and memory and are thought to be functionally analogous to the vertebrate hippocampus [29,30]. Expression of the randomly inserted hTau^0N4R^ and hTau^2N4R^ during development under elavG4 impaired the late-dividing mushroom body neuroblasts, resulting in aberrant and reduced in-size mushroom body neurons (MBNs), while other neuropils such as the protocerebral bridge remained unaffected [21,24]. In fact, while hTau^0N4R^ and hTau^2N4R^ expression impaired MBN development, hTau^0N3R^ did not, despite its relatively elevated amount relative to that of the former isoforms [21,24]. However, unlike for the randomly inserted hTau^0N4R^, pan-neuronal expression of the same transgene integrated into a different chromosomal site did not affect MB structure [26]. As the MB structural deficits correlate with hTau levels in early embryogenesis [24], we hypothesized that the insulated from position effects attP landing sites attenuate early embryonic expression of the inserted tau transgenes [26], resulting in grossly normal MBs. To address this hypothesis systematically, we used this new set of hTau isoforms at the same landing site, facilitated by the quantification of their protein levels (Figure 1).

As shown in Figure 2A for the posterior of the brain, the overall structure and morphology of the CNS appeared unaltered. However, upon close examination, the MBs of animals expressing particular isoforms appeared reduced in size. To quantify this potential difference, the area covered by the dendritic fields of MBNs, known as calyces, was carefully estimated. This revealed small but statistically significant differences in the size of the calyx in animals expressing hTau^0N3R^, hTau^1N3R^, hTau^2N3R^ and hTau^0N4R^ (Figure 2A bottom and Appendix A). Notably, there does not appear to be a correlation between reduced calyx size and hTau isoform accumulation in the CNS (Figure 1A). One of the most abundantly accumulating isoforms, hTau^1N4R^ and one of the least abundant, hTau^2N4R^, appear not to precipitate statistically significant size reductions (Figure 2A bottom and Appendix A). Conversely, accumulation of the low-abundance hTau^2N3R^ and the more abundant hTau^1N3R^ results in reduced calyx size. However, all 3R isoforms and the smaller of the 4R, hTau^0N4R^, reduce the size of the MBs (Appendix A). It should also be noted that hTau^0N3R^ accumulation results in this small reduction in MB size but does not affect its overall structure. This likely accounts for the reports [21,24] indicating that the randomly inserted hTau^0N3R^ does not alter MB structure as these studies concentrated on gross morphological defects and did not use the careful quantitation method used herein.


**4R hTau isoform accumulation affects associative learning and memory**


In agreement with the early appearance of cognitive deficits in human tauopathy patients, associative learning and memory are impaired in Drosophila tauopathy models [21,22,24,26,27,31,32,33]. Given that the MBs are cardinal for these behavioral outputs [29,30] and their size is differentially affected by particular hTau isoforms, we aimed to systematically examine the consequence of all hTau isoforms in associative learning and memory and whether any defects correlate with protein levels, or MB size differences.

As before, the well-established Pavlovian olfactory conditioning paradigm [34] was used, which requires the flies to associate one odorant with aversive electric foot shocks and avoid it preferentially. To increase the resolution of the assay, three and six odor/footshock associations were used for conditioning and performance was assessed immediately after (learning, or 3-min memory). Significantly, the accumulation of all 4R isoforms resulted in highly significant (Appendix A) learning impairment both at three and six stimulus pairings (Figure 2B). In contrast, animals accumulating 3R isoforms performed normally, except for those expressing hTau^0N3R^. These flies presented defective learning after 3 stimulus pairings but not after six (Figure 2B). This suggests that hTau^0N3R^- expressing flies are defective in the rate of learning and not learning *per se*, as increased pairings ameliorate the deficit. This effect of hTau^0N3R^ cannot be attributed to its expression, as its levels are not significantly different from the other 3R isoforms (Appendix A).

Does accumulation of hTau isoforms affect the memory of the odor/footshock association differentially? Two forms of consolidated memory can be elicited in Drosophila by repeated cycles of 12 odor/footshock pairings and assessed 24 h later. A protein synthesis-dependent memory (PSDM) was induced by 5 cycles of pairings spaced 15 min apart, while a protein synthesis independent memory (PSIM) was elicited by the same number of conditioning cycles but without the intervening rest interval [35,36].

In agreement with the learning results, expression of all 4R isoforms resulted in significantly impaired PSDM, while accumulation of 3R isoforms did not affect this form of consolidated memory (Figure 3A and Appendix A). In contrast, PSIM was not affected by the accumulation of any hTau isoform (Figure 3B and Appendix A). These results are in agreement with prior reports [22,32,33] and expand them to include all hTau isoforms. Importantly, the specificity of consolidated memory defects to PSDM for all 4R isoforms is consistent with the interpretation [37] that hTau excess of these hTau species in the fly CNS impairs translation-related processes required for PSDM [29,36]. Furthermore, these results strongly support the notion that the severity of learning and memory impairment is not proportional to hTau levels (Figure 1) or the mild reduction in MB size (Figure 2A), but rather correlates with the number of microtubule-binding repeats.

To ascertain that the learning and memory deficits are not a consequence of reduced olfactory and mechanosensory acuity due to hTau excess, the naïve responses to these stimuli were quantified for animals expressing all isoforms (Figure 4A and Appendix A). Controls and hTau-expressing flies avoided equally the aversive odors of benzaldehyde and 3-octanol, as well as the 90-V electric footshock. Therefore, expression of all hTau isoforms in the fly CNS does not affect perception and reactivity to the aversive stimuli used for conditioning and does not account for the defects uncovered.

As an added pertinent control, since flies have to move to different arms of the choice maze and impaired movement might appear as reduced cognitive performance, locomotion was assessed in hTau-expressing animals. To increase resolution, the task was made more demanding by assessing innate locomotor behavior based on negative geotaxis (climbing). When tapped to the bottom of a vial, flies will attempt to escape by climbing to the top, and the assay has been used to demonstrate locomotor dysfunction in aged tau-expressing animals [22,38,39]. As shown in Figure 4B, the climbing ability of the relatively young *hTau*-expressing animals used in all our assays was not significantly different from controls (Appendix A). Collectively, these results indicate that pan-neuronal hTau accumulation does not precipitate deficits in sensory modalities requisite for olfactory conditioning.


**Differential effects of hTau isoforms on footshock habituation.**


In addition to their established function in olfactory learning and memory, MBs also play a role in the habituation of repeated footshocks [37,40,41]. Habituation is a form of non-associative plasticity manifested as a reduction in response to specific inconsequential repetitive stimuli. The endogenous Drosophila tau (dTau) functions in habituation since dTau null mutants fail to habituate to footshocks as controls do, whereas dTau overexpression results in premature habituation [37]. Because footshock habituation is sensitive to tau levels, we hypothesized that hTau could also affect this non-associative process as overexpression of dTau does [37]. Hence, flies expressing pan-neuronally the six hTau isoforms were subjected to the standard habituation assay of repeated 45V footshocks [40,41].

As shown in Figure 5A, controls habituated normally to 15 footshocks, as did flies expressing all hTau isoforms except hTau^1N3R^ (Appendix A). Importantly, the habituated response of hTau^2N3R^, hTau^0N4R^ and hTau^1N4R^-expressing flies was actually premature as it occurred after only two stimuli, which do not suffice for control animals to habituate to (Figure 5B and Appendix A). Consistently, hTau^1N3R^-expressing flies did not habituate prematurely (Figure 5B). Therefore, it appears that excess hTau^1N3R^ inhibits processes necessary for habituation, while hTau^2N3R^, hTau^0N4R^ and hTau^1N4R^ promote it. Whether these effects are mediated by the same or distinct neuronal circuits in the fly CNS is part of ongoing research, but these results clearly differentiate the roles of these hTau isoforms in the process. Additional support for the specificity of the effects is provided by the lack of habituation aberrations in flies expressing hTau^0N3R^ and hTau^2N4R^, both of which affect associative learning and memory (Figure 2B and Figure 3A), despite their lower accumulation levels (Figure 1A).


**All hTau isoforms alter circadian activity.**


Circadian behavioral deficits, such as increased night-time wakefulness and increased daytime sleep, are common in Alzheimer’s disease and related tauopathies [42]. Flies exhibit daily cycles of activity and inactivity, a behavioral rhythm that is governed by the animal’s endogenous circadian system [43]. To investigate how the six hTau isoforms affect circadian behavior, we expressed them pan-neuronally and monitored locomotor activity under a 12 h light-dark (LD) cycle. As expected, the activity profiles contained morning and evening activity peaks, centered around the light transitions (lights on and off), separated by a midday siesta and a period of sleep during the night (Figure 6 left panel).

These two periods of activity are controlled by the endogenous clock, and alterations in their timing may indicate a change in the circadian behavior. However, the rhythmicity of these activity peaks in hTau-expressing flies was maintained as for controls for the two 24-h periods the activity was monitored (Figure 6 left panel). Another property indicative of proper clock function is the anticipatory increase in locomotor activity, which occurs prior to the dark-to-light or light-to-dark transition. This was assessed by quantification of the total activity of all flies monitored over the two days and displayed per 6 h (Figure 6 right panel). Interestingly, all hTau-expressing flies presented significantly increased activity levels compared to controls (Appendix A), largely during the evening light-to-dark transition. Notably, flies over-expressing hTau^0N3R^, hTau^0N4R^ and hTau^1N4R^ also showed statistically significantly elevated activity during the dark-to-light transition in the morning (Figure 6 right panel and Appendix A).

Collectively, these results indicate that expression of hTau^0N3R^, hTau^0N4R^ and hTau^1N4R^ in the fly CNS results in significant increases in activity at the light transition periods, while in flies expressing hTau^1N3R^, hTau^2N3R^ and hTau^2N4R^, elevated activity is limited to the evening light-to-dark transition. It seems that the more abundantly accumulating species correlate with the higher overall light transition activity, with the exception of hTau^0N3R^, which elevates activity at both transition periods but whose levels do not appear particularly high. Therefore, hTau levels do not appear to affect the activity of flies expressing them as much as the particular isoforms do.

**Isoform-specific elevation of oxidative stress vulnerability**.

Vulnerability to oxidative stress has been employed as a measure of tau accumulation-dependent toxicity in tauopathies including Alzheimer’s disease [44,45] and has been modeled in Drosophila [32,33,46,47]. Hence, the resistance of adult hTau expressing flies and controls to 5% H_2_O_2_ was assessed as before [32,33,48].

Mortality of hTau-expressing flies was significantly increased as early as 24 h of exposure to 5% H_2_O_2_ and presented a steady significant increase over controls over the duration of the experiment (Figure 7, and Appendix A). Importantly, flies expressing the three 4R isoforms presented significantly enhanced susceptibility to oxidative injury as they started to expire much earlier than those expressing 3R isoforms (Figure 7, 24 h and 48 h and Appendix A). Moreover, hTau^2N3R^- expressing animals appeared more vulnerable to H_2_O_2_ than those expressing hTau^1N3R^ (Figure 7, 52 h and 58 h and Appendix A). In contrast, flies expressing the hTau^0N3R^ isoform were the most resistant since compared to controls, they presented statistically significantly higher mortality only upon treatment for 75 h (Figure 7 and Appendix A, 75 h). Clearly, therefore, vulnerability to H_2_O_2_ toxicity correlates well with the number of microtubule-binding repeats, but not with protein abundance in the fly CNS.


**Adult lifespan is equally reduced by all hTau isoforms.**


An ultimate test of hTau toxicity, which also affords resolution of potential differences, is the survival of animals expressing pan-neuronally hTau proteins [26,27]. To ascertain that any differences uncovered would not be attributed to isoform-specific developmental compromises [24], the inducible TARGET system [49] was utilized, which permits pan-neuronal transgene expression only in adulthood. To that end, animals were raised at 18 °C and when adult, they were transferred and maintained at 30 °C to induce expression from the hTau transgenes [27].

The survival of these hTau isoform-expressing flies was monitored until they all expired. As demonstrated in Figure 8, all hTau isoforms presented statistically significant shortened lifespans compared to controls (Appendix A). This is evident both in maximal survival, which was 35 days for controls and 31–32 for hTau-expressing flies and their 50% attrition date, which was day 25 for hTau-expressing flies and day 27 for controls (Figure 8). The attrition measure refers to the date when 50% of the population of a given genotype expires and is presented at day 27 in the inserts in Figure 8 for clarity. Clearly, the expression of all hTau isoforms results in a statistically significant survival decrease on this day (Appendix A). These results are congruent with those of Fernius [25], who concentrated only on the maximal survival day and expanded them by incorporating survival measures at the 50% attrition day, which offers an increased resolution of the toxicity phenotype. Importantly, there does not appear to be a correlation between the premature mortality of hTau-expressing animals and the level of the particular isoform accumulating in their CNS, indicating equal toxicity for all 3R and 4R isoforms.

## 3. Discussion

Animal and cellular models of tauopathies aim to mimic human disease symptoms and pathology [50,51] towards understanding mechanisms of pathogenesis mediated by hTau deregulation. A plethora of such models are in use today [50,51] and are characterized by the expression of a single hTau isoform as a representative mediator of pathology [52,53]. The collective contribution of all animal and cellular models to our current understanding of tauopathy pathogenesis mechanisms is invaluable [54]. In patients, however, multiple isoforms are involved in some tauopathies such as Alzheimer’s disease, while Pick’s disease apparently involves only 3R species and the 3R/4R ratio alteration appears to characterize most of these disorders [7]. The contribution of each hTau isoform involved in the pathogenesis and pathology of each distinct tauopathy is currently unclear. Although still isolated from the rest, an essential step forward toward that goal is to determine the contribution of each isoform to typical tauopathy presentations.

Initial answers to this question are presented above and summarized in Figure 9. Despite the differences in steady-state levels among the isoforms in the fly CNS, it appears that the smaller isoforms affect similarly the size of the MB calyx, most likely a consequence of impaired MB neuroblast division [24]. This phenotype differs somewhat from previous work in that randomly inserted hTau^2N4R^ was shown to affect MB structure significantly and hTau^0N3R^ not to [24]. This could be a consequence of increased protein levels in the case of the randomly inserted transgenic protein, which are rather low for the hTau^2N4R^ protein used herein (Figure 1A). In the case of the randomly inserted hTau^0N3R^, potential mild defects could have evaded detection then [24], which are uncovered herein because of the precise measurement method employed (Figure 1A).

However, all isoforms decrease longevity more or less equally, despite their differences in steady-state levels. Vulnerability to oxidative stress likely contributes to this premature mortality, with an accumulation of 4R isoforms being more potent mediators than their 3R counterparts. Similarly, all hTau isoforms elevate locomotor activity, apparently in anticipation of subjective dawn and dusk, potentially a manifestation of anxiety and restlessness, also observed in patients [55,56].

Significantly, associative learning and protein synthesis-dependent memory are compromised only by the accumulation of 4R but not 3R isoforms. This suggests that 4R hTau isoforms might interfere with or inhibit regulated translation, which underlies this form of memory [36]. It is unclear at the moment whether the preferential interference of 4R isoforms with PSDM reflects their apparent more effective binding to microtubules, potentially impeding processes requiring more plasticity. A role for the endogenous Drosophila tau in translation regulation and PSDM formation has been reported recently [37], in congruence with data from vertebrate systems [57]. Interestingly, Pick’s disease involving largely 3R isoforms does not present significant learning and memory deficits but other dementia manifestations [58,59]. We have not yet tested compulsive behaviors such as persistent grooming in Drosophila, which characterize Pick’s disease and other such dementias and will strengthen this still tenuous correlation.

Altered habituation upon hTau accumulation suggests putative roles for particular isoforms in fly CNS synapses, in agreement with reports from other systems [60,61]. Regulated neurotransmission is thought to underlie habituation [62] and, interestingly, four hTau isoforms affect this process. As summarized in Figure 9, hTau^1N3R^ impedes habituation, whereas hTau^2N3R^, hTau^0N4R^ and hTau^1N4R^ lead to premature habituation, a manifestation of more expedient devaluation of recurrent stimuli [40]. As inhibition of premature habituation and normal habituation onset require neurotransmission from different types of MB neurons [40,41], it is possible that hTau^1N3R^ and the other three hTau isoforms affect neurotransmission from different neuronal types. This hypothesis, currently under investigation, might provide additional support in favor of the differential functional specificity of hTau isoforms.

A consistent conclusion from the work presented above is the lack of correlation in isoform abundance with the effects mediated by hTau isoform accumulation in the fly CNS. Therefore, the specific defects elicited by hTau isoforms are most likely a consequence of their structural elements-mediated functional specialization and differential interactions with intracellular partners, possibly in a neuronal type of specific manner.

It is, however, of interest to determine the cause of the apparent differential stability of the transgenic hTau proteins in the Drosophila CNS. Their steady-state levels do not correlate with their size, N or R repeats, suggesting that the instability is not mediated by primary sequence elements. However, potential secondary and tertiary structures could in principle differ among isoforms, rendering them differentially stable or unstable in the fly CNS. An alternative explanation currently under investigation is that tau isoforms may be stabilized by interactions with other tau species. If so, then hTau^1N3R^, hTau^0N4R^ and hTau^1N4R^ could be stabilized by the endogenous dTau, whereas interaction with hTau^0N3R^, hTau^2N3R^ and hTau^2N4R^ might be significantly weaker or not possible at all. The availability of a null dTau strain enables the formal address of this hypothesis. Moreover, experiments utilizing co-expression of hTau species aiming to equalize their levels in the presence or absence of dTau will further elucidate the issue.

Understanding tauopathies requires comprehension of the role and potential contribution of each hTau species in their characteristic pathologies within tissues and cell types, whether they are expressed alone or with their companion isoforms. We have revealed a number of common and hTau isoform-specific effects in the Drosophila model (Figure 9). Namely, all isoforms affect locomotor activity, oxidative stress response and survival. On the other hand, the 4R isoforms alone exhibit functions related to learning and memory, while only 0N4R and 1N4R, along with 1N3R and 2N3R, appear to be involved in habituation. Moreover, despite their effect on cognitive functions, 1N4R and 2N4R are the only isoforms not associated with MB size regulation. Everything considered, each hTau isoform has a unique functional profile. The genetic arsenal and facility of the fly model system will likely enable us to expand the results of this initial study, provide answers to questions posed above and facilitate our understanding of hTau physiology and pathobiology.

## 4. Materials and Methods


**Drosophila culture and strains:**


Flies were cultured in standard sugar-wheat flour food supplemented with soy flour and CaCl_2_ [40]. Crosses, unless otherwise stated, were kept at 25 °C and 70% relative humidity with a 12 h light-dark cycle. For pan-neuronal transgene expression the *elav^C155^*-GAL4 [63] was employed. The *elav^C155^*-GAL4; *tub-Gal80ts* strain was constructed using standard methods [49]. The fly lines carrying the six UAS-hTau isoforms were kindly provided by Dr. Stefan Thor (Linkoping University, [25]) and were backcrossed into the resident Cantonized *w^1118^* control background for five generations.


**Molecular and histological:**


RNA extraction and RT-PCR: As previously described [32], total RNA was extracted from 20 Drosophila heads (1 to 3 days post-eclosion) using TRI Reagent^®^ (Sigma-Aldrich, St. Louis, Mo., USA), following the manufacturer’s instructions. Reverse transcription from DNase I treated total RNA was carried out using SuperScript^®^ II Reverse Trancriptase (Invitrogen, Waltham, MA, USA) and cDNA was subjected to PCR using the Go Taq^®^ Flexi DNA Polymerase (Promega, Madison, WI, USA). The ribosomal gene *rp49* was used as a standardizing control of the RT. Four independent biological repeats were performed. Quantification was performed using ImageJ (NIH) and the resultant means were compared with the least square means planned comparisons method following an initial positive ANOVA.

Western blotting and Antibodies: Seven fly heads at 1–3 days post-eclosion were homogenized in 1x Laemmli buffer (50 mM Tris pH 6.8, 100 mM DTT, 5% 2-mercaptoethanol, 2% SDS, 10% glycerol and 0.01% bromophenol blue), boiled and separated in 10% SDS-acrylamide gels. Proteins were transferred to PVDF membranes and probed with mouse monoclonal anti-tau (5A6, Developmental Studies Hybridoma Bank) at 1:1000 dilution. Membranes were concurrently probed with an anti-syntaxin primary antibody (8C3, Developmental Studies Hybridoma Bank) at a 1:3000 dilution. The anti-mouse HRP-conjugated secondary antibody was applied at 1:5000 dilution and proteins were visualized with chemiluminescence (Immobilon Crescendo, Millipore). Signals were quantified by densitometry with the Image Lab 5.2 program (BioRad) and results were plotted as means ± standard error of the mean (SEM) from three independent experiments. The means were compared using the least square means method following an initial positive ANOVA.

Immunohistochemistry: Adult female flies at 2–3 days old were fixed in Carnoy’s solution (60% ethanol, 30% chloroform, 10% glacial acetic acid), and embedded in paraffin. Serial frontal sections (4 μm) were prepared through the entire fly brain. Slides were processed as previously described [24] through xylene, ethanol, water and into PBHT buffer (0.01 M NaH_2_PO_4_, 0.25 M NaCl, 0.2% Triton X-100, pH 7.4). Slides were blocked in PBHT containing 10% normal goat serum for 3 h at room temperature. Immunostaining was performed using rabbit anti-LEO [64] at 1:4000. Images were acquired at 40× with an Olympus BX53 microscope. The area of mushroom body calyces was estimated using Image J (NIH). Results were plotted as means ± SEM from at least seven independent experiments. The data were analyzed by Dunnett’s tests relative to control driver heterozygotes following an initial positive ANOVA.


**Neuronal function and toxicity assays:**


Husbandry: Mixed-sex populations of from 2 to 5-day-old flies were used for all behavioral experiments. UAS-hTau transgenic strain males were crossed *en masse* with *Elav**^C155^*-Gal4 driver females at 25 °C. Upon eclosion they were separated in groups of 50–70 animals in vials and placed at 30 °C overnight to boost transgene expression.

Associative learning and memory: Flies were trained with classical olfactory aversive conditioning protocols [34] as previously described [22,37]. We used benzaldehyde (5% *v*/*v*) and 3-octanol (50% *v*/*v*) diluted in isopropylmyristate (Fluka) as standard odorants. Training and testing were carried out at 25 °C and 70–75% relative humidity under dim red light. To assess learning, flies were tested immediately after a single training cycle consisting of 15 or 30 s odor A with either three or six 90 V footshocks at 4.5 s inter-stimulus intervals, 30 s room air and 15 or 30 s odor B without reinforcement. For protein synthesis-dependent memory (PSDM), flies underwent 5 training cycles spaced 15 min apart and tested 24 h later. For protein synthesis independent memory (PSIM), flies were also submitted to five conditioning cycles, but without the 15 min rest interval (massed conditioning) and memory was tested after 24 h. In each cycle, flies were exposed for 1 min to odor A paired with twelve 90 V electric shocks at 4.5 s inter-stimulus intervals, followed by 30 s of air and 1 min odor B without reinforcement. The performance index was calculated by subtracting the fraction of flies in the shock-associated odorant arm of the maze from those in the opposite arm and averaging the outcome from the two mazes where the complementary odorants were associated with the odor as described before [22,37].

Electric footshock and odor avoidance: Experiments were performed as described before [65]. Briefly, ∼70 flies were placed in a T-maze and allowed to choose for 90 s between an electrified (90 V shocks every 4.5 s) and an otherwise identical inert standard copper grid. Odor avoidance was quantified by exposing flies to an airstream carrying the odor in one arm of a T-maze and fresh air in the other. Flies were given 90 s to choose between aversive odors and air. The odorants utilized for these experiments were as described for associative learning and memory assays. At the end of the choice period, flies in each arm were trapped and counted. The performance index (PI) was calculated as the percentage of the fraction of flies that avoid the odor/shock minus the fraction of flies that prefer the odor/shock-bearing arm.

Climbing assay: Approximately two days after eclosion, male flies were collected in groups of 10–12 and aged for 5 days at 25 °C. Each group of flies was transferred in an empty vial (2 cm diameter) at 25 °C, 40% relative humidity under dim red light and was allowed 1 min to acclimate. The vial was divided into three compartments of 2 cm each (bottom, middle, upper). Flies were gently tapped to the bottom of the vial, and their negative geotaxis reflex response was recorded by camera (Basler AG acA1920–155 μm, Germany) with rate of 1 frame per second (FPS). To determine the optimal frame to analyze, the time required for control flies to ascent to the top was estimated and a standard curve was constructed (Appendix A) demonstrating that 3 s after initializing the response affords the most resolution as only 50% of the flies are in the top. Consequently 3 s after taping the flies to the bottom of the vial animals in each compartment were counted and compartment frequency was calculated as the quotient of flies in every compartment divided by the total number in the group. Comparison of the climbing ability was performed using the least squares means (LSM) approach relative to control driver heterozygotes.

Habituation to electric footshock: Experiments were performed as described before [41,65]. The avoiding fraction (AF) was calculated as described [41], by dividing the number of flies avoiding 45 V shock by the total number of flies. For the training phase ∼70 flies were exposed to either 15 or 2 electric shocks at 45 V. After a 30 s rest, flies were tested by choosing for 90 s between an electrified (45 V) and an inert grid. At the end of the choice period, the flies in each arm were trapped and counted and the habituation fraction (HF) was calculated by dividing the number of flies preferring the electrified grid by the total number of flies. Finally, the habituation index (HI) was calculated by subtracting the avoidance fraction from the habituation fraction multiplied by 100. HI represents the change in footshock avoidance contingent upon prior footshock experience (habituation).

*Statistical analysis of behavioral experiments*: All genotypes involved in an experiment were tested per day and data were analyzed parametrically with the JMP statistical package (SAS Institute Inc., Cary, NC, USA) as described before [22,37]. Performance indices calculated for each genotype were examined for differences using ANOVA, followed by planned multiple comparisons using the least squares means (LSM) approach.

Locomotor activity: Locomotor activity was recorded with the TriKinetics DAM2 monitors, which use infrared beams to automatically detect and record locomotor activity of individual flies. Briefly, male flies (2–5 days old) were introduced into transparent locomotor activity tubes (5 mm diameter) with agar food supplemented with fructose and sugar (1.5% *w*/*w* fructose, 3% *w*/*w* brown sugar, 0.25% nipagen and 1.5% agar) at one end and cotton plug at the other. Flies were transferred to locomotor activity behavior for at least 12 h for acclimatization before collecting data and then monitored for about 2 days at 25 °C in a 12 h light/dark cycle. The average of raw activities per fly for 30-min bins were collected and summed into four 6-h intervals per day (early day: 600–1130, late day: 1200–1730, early night: 1800–2330 and late night 2400–530 h). Activity data of flies that died within 12 h of recording were not considered. Averaged activities were compared to that of designated control using the least squares means (LSM) approach.

H_2_O_2_ sensitivity: Animals accumulating the six human tau isoforms under the control of the pan-neuronal *elav^C155^*-GAL4 driver were raised at 25 °C together with control driver heterozygotes. Oxidative stress test was performed at 25 °C using a medium containing 10% sucrose, 1xPBS, 0.8% low melt agarose and 5% H_2_O_2_. Twenty young males (1–3 days old) were transferred to fresh medium every 24 h [48]. At least 300 flies were assessed per genotype. Means of survival were compared to that of designated control using the least squares means (LSM) approach.

Lifespan Determination: Animals accumulating the six hTau transgenes under *elav^C155^*-GAL4; *tub-Gal80ts* were raised at 18 °C together with control driver heterozygotes. Groups of 20 young male flies (1–3 days old) were collected and maintained at 30 °C until they expired. Flies were transferred to fresh vials every 2 days. At least 300 flies were assessed per genotype. Survival curves were compared using log-rank tests (JMP 7.1 statistical software package, SAS Institute Inc.).

## Figures and Tables

**Figure 1 ijms-23-12985-f001:**
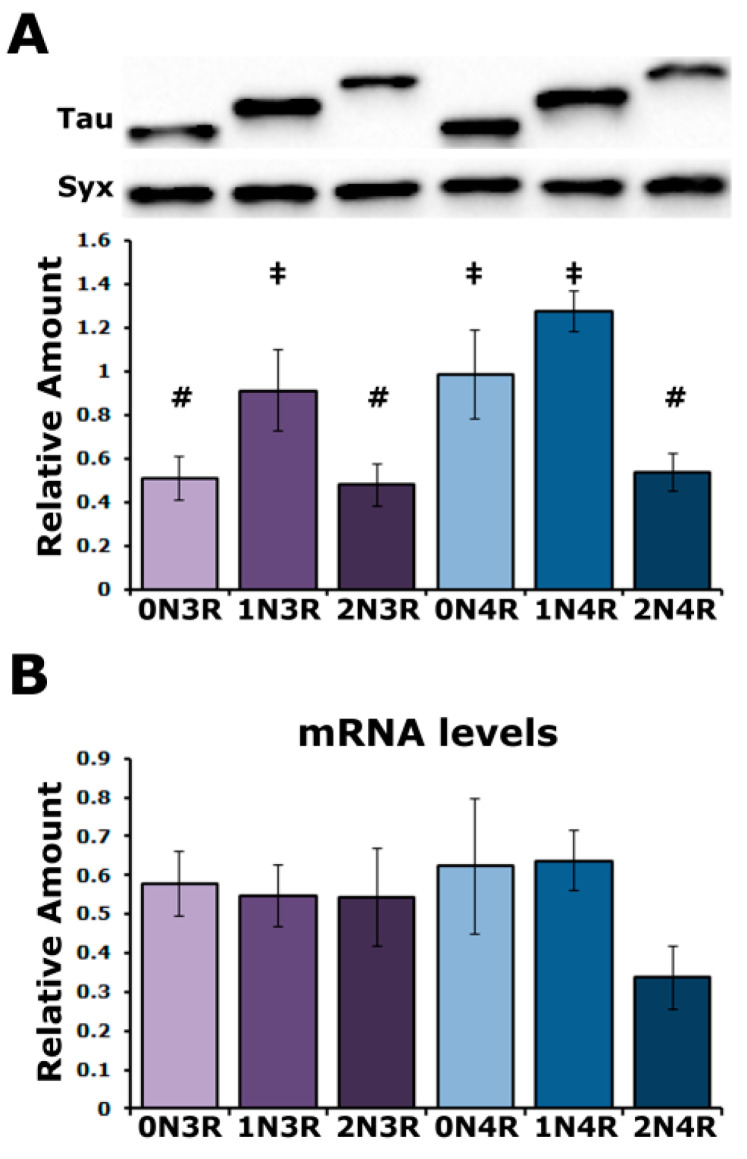
**Variable levels of hTau isoforms in the Drosophila CNS.** (**A**) A representative Western blot of head lysates from animals expressing the different hTau transgenes under the elavGAL4 driver detected with the 5A6 anti-tau antibody. The hTau isoform expressed in the fly CNS is indicated below the quantification. For the quantification, hTau levels were normalized using the Syx loading control and are shown as the mean ± SEM of n = 3 independent experiments. Bars marked with # are not significantly different from each other but are from those marked with ‡ and conversely ‡ marked bars are not significantly different from each other but are from ones marked with #. (**B**) Quantification of tau mRNA levels by Reverse Transcription followed by the Polymerase Chain Reaction in the CNS of flies expressing the indicated hTau transgenes under elavGAL4. The *rp49* RNA served as an internal reference transcript for the reaction and has been used to normalize the quantifications. Bars indicate mean ± SEM relative mRNA levels. n = 4. Statistical details for both experiments are presented in Appendix A.

**Figure 2 ijms-23-12985-f002:**
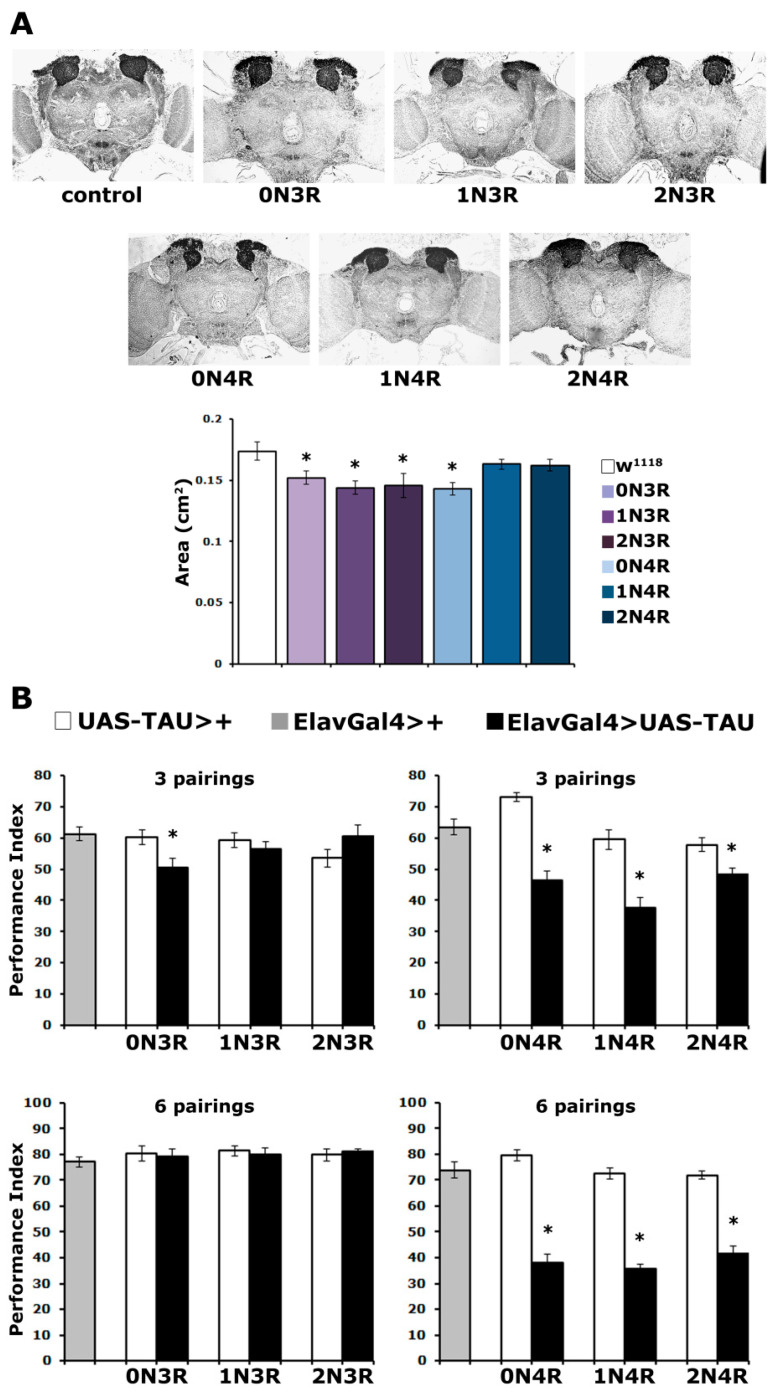
**Tau isoforms affect differentially mushroom body neurons structurally and functionally.** (**A**) Carnoy’s-fixed paraffin-embedded frontal sections stained with anti-Leonardo antibody at the level of the dendrites (calyces) of MB neurons, from control (elavGAL4/+) and animals expressing the indicated hTau transgenes under elavGal4. The area of the calyx from multiple similar sections per genotype was quantified, averaged below and presented as the mean ± SEM. Stars indicate significant differences from control flies. n ≥ 7 for all genotypes. (**B**) Learning after 3 and 6 pairings of animals accumulating pan-neuronally the indicated hTau transgenes under elavGAL4 (black bars) compared with driver (grey bars) and transgene heterozygotes (white bars). The means ± SEMs are shown. Stars (*) indicate significant differences from both controls. n ≥ 11 for all genotypes. Statistical details for both experiments are presented in Appendix A.

**Figure 3 ijms-23-12985-f003:**
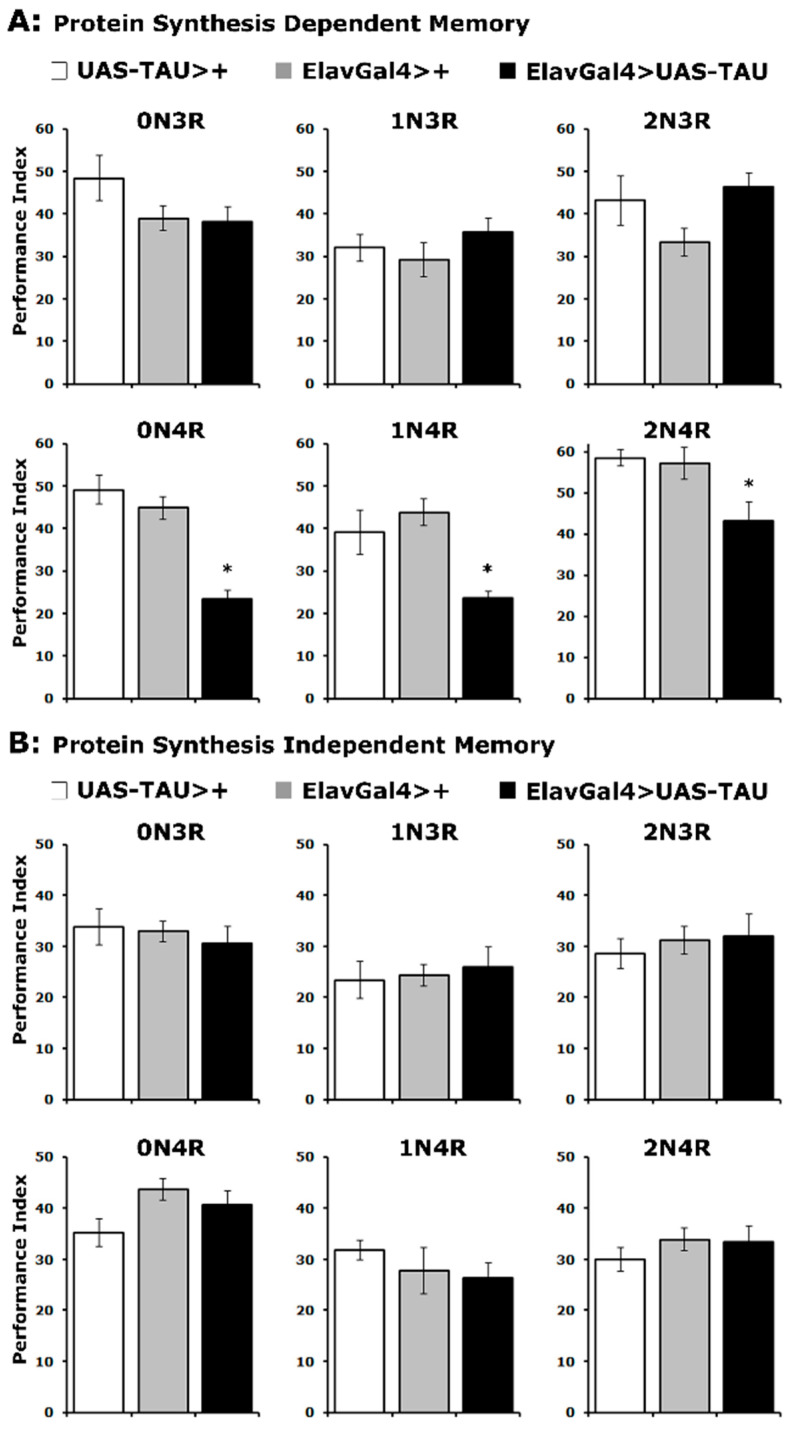
**PSD memory deficits emerge for all 4R isoforms.** The hTau isoform expressed is indicated above each graph. The means ± SEMs for the indicated repetitions are shown. Stars (*) indicate significant differences from both controls. Statistical details in Appendix A. (**A**) Protein synthesis-dependent memory of animals accumulating pan-neuronally the indicated hTau isoforms (black bars) compared with driver and transgene heterozygotes (grey and white bars). n ≥ 7 for all genotypes. (**B**) Protein synthesis independent memory of animals accumulating pan-neuronally the indicated hTau proteins (black bars) compared with driver and transgene heterozygotes (grey and white bars). n ≥ 9 for all genotypes.

**Figure 4 ijms-23-12985-f004:**
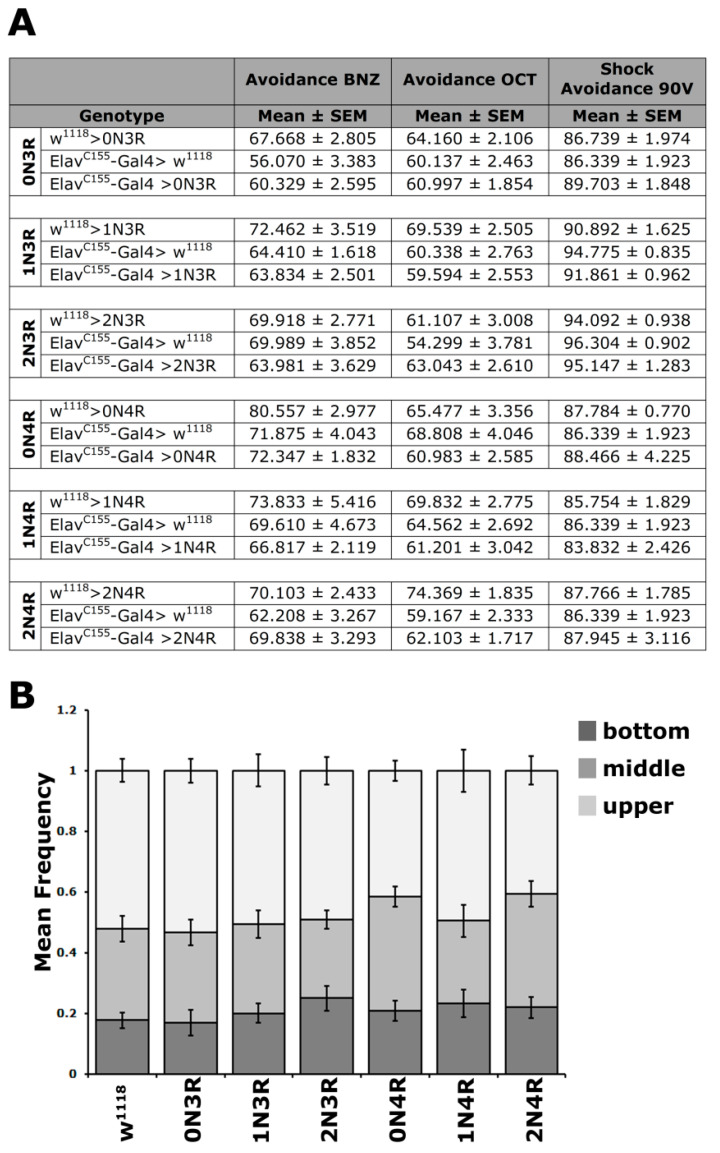
**Conditioning stimulus perception and mobility is unaffected by hTau isoform accumulation.** (**A**) Odor and electric footshock avoidance of animals accumulating pan-neuronally the indicated hTau isoforms compared with driver and transgene heterozygotes. The means ± SEM are shown for n ≥ 6 for all genotypes. (**B**) Negative geotaxis (climbing) of flies accumulating pan-neuronally for 5 days at 25 °C the indicated hTau isoforms compared with driver heterozygotes. The hTau isoform expressed is indicated below each bar. The means ± SEMs are shown for n ≥ 12 for all genotypes. Statistical details for both experiments are presented in Appendix A.

**Figure 5 ijms-23-12985-f005:**
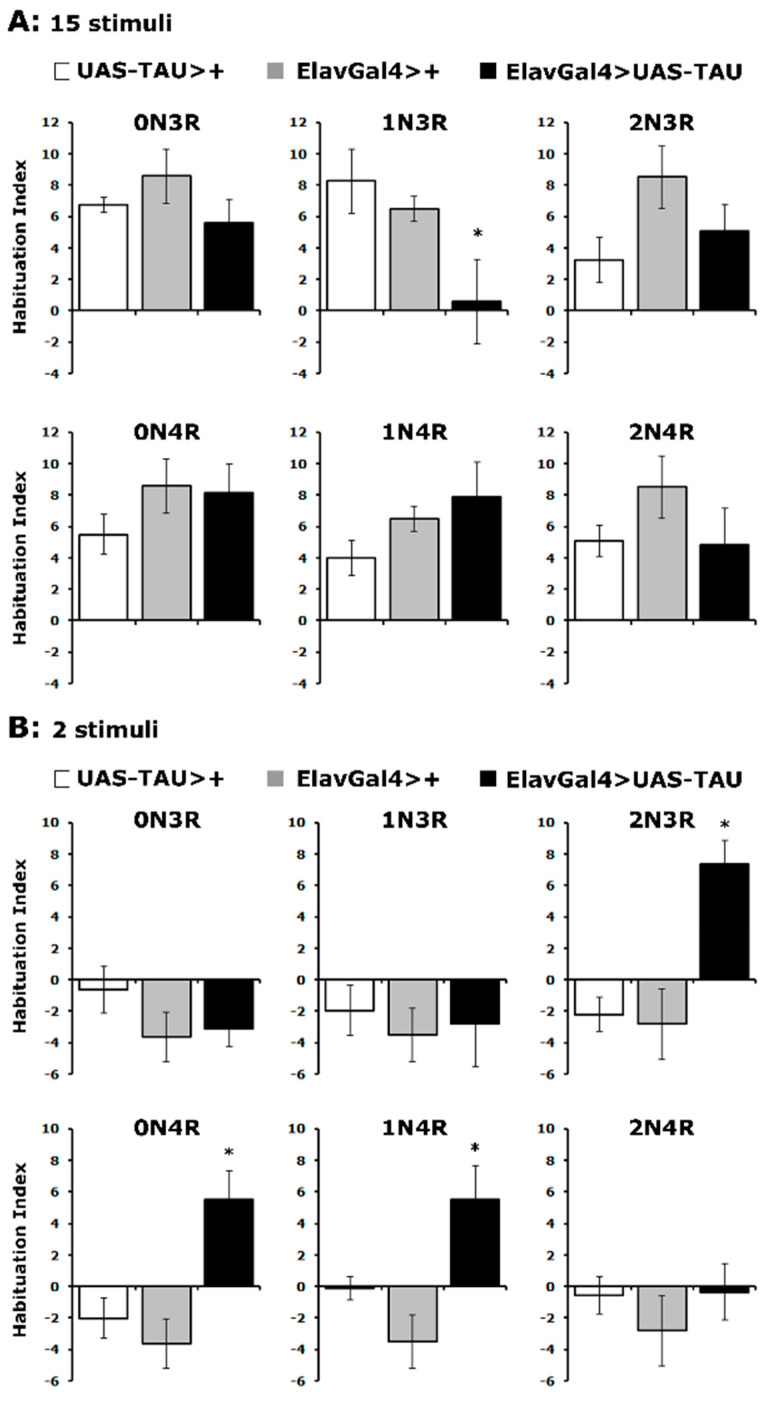
**hTau isoforms affect differentially habituation to footshocks.** Habituation following exposure to 15 (**A**) or 2 footshocks (**B**) of flies accumulating pan-neuronally the indicated hTau isoforms (black bars) compared with driver and transgene heterozygotes (grey and white bars). The means ± SEMs are shown for n ≥ 7 for all genotypes. Stars (*) indicate significant differences from both controls. Statistical details in Appendix A.

**Figure 6 ijms-23-12985-f006:**
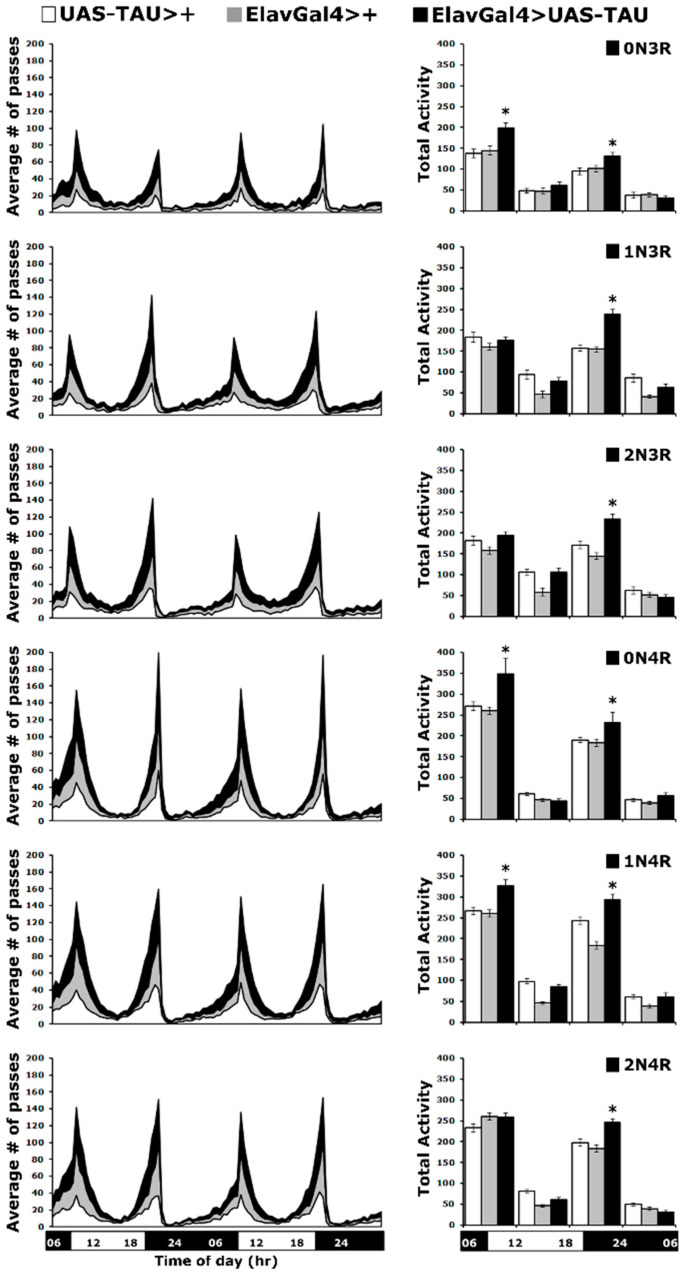
**Circadian locomotor activity is affected by all hTau isoforms.** Locomotor activity of animals accumulating pan-neuronally the indicated hTau isoforms under elavGAL4 (black bars) compared with driver and transgene heterozygotes (grey and white bars). Flies were monitored for 2 days at 25 °C in a 12 h light/dark cycle. Representations of the average activities of the indicated genotypes monitored in 30-min bins over two days (left panel) divided in four 6-h intervals (early day: 0600–1130, late day: 1200–1730, early night: 1800–2330 and late night 2400–0530 h) as indicted by the light on (white) and off (black) and bar on the bottom of the graph. Total activities shown as means ± SEMs for each quarter day for animals accumulating the indicated isoform and relevant controls are indicated on the right. Stars (*) indicate significant differences from both controls. n ≥ 50 flies per genotype. Statistical details in Appendix A.

**Figure 7 ijms-23-12985-f007:**
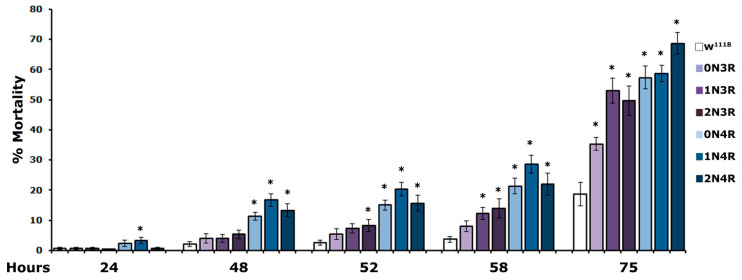
**CNS accumulation of all hTau isoforms increases vulnerability to oxidative stress.** The mortality of flies accumulating the indicated hTau isoforms and controls upon exposure to 5% H_2_O_2_ at 25 °C scored at 24, 48, 52, 58 and 75 h is shown. The bars represent the mean ± SEM from two independent experiments with at least 300 flies assessed per genotype. Stars indicate significant differences from control flies. Statistical details in Appendix A.

**Figure 8 ijms-23-12985-f008:**
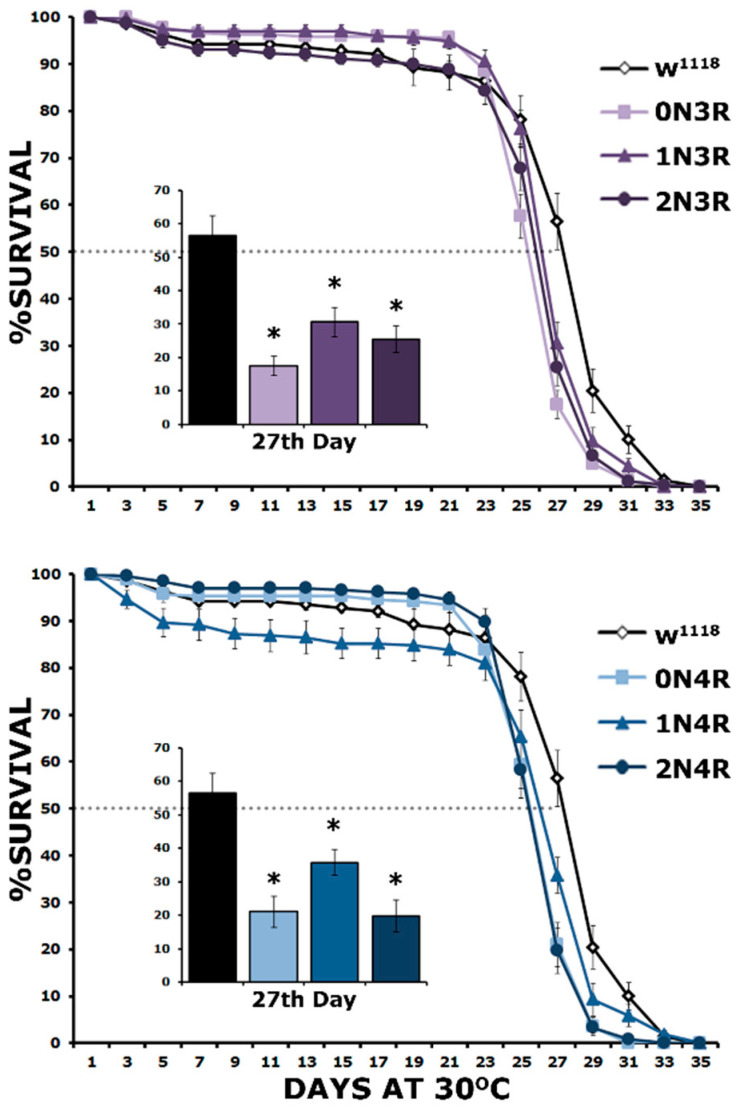
**Lifespan is equally decreased by all hTau isoforms.** Survival curves for animals accumulating pan-neuronally the six hTau isoforms under elav-GAL4; tub-Gal80ts compared with driver heterozygotes (w^1118^). Flies were raised at 18 °C, but adults were transferred and maintained at 30 °C until they expired. The data represent the mean ± SEM from two independent experiments with at least 300 flies assessed per genotype. After 27 days, the population of control heterozygotes (w^1118^) was reduced by 50% (50% attrition) indicated by the dotted line and the number of surviving animals per genotype on that day is quantified in the insert. Stars indicate significant differences from control. Statistical details in Appendix A.

**Figure 9 ijms-23-12985-f009:**
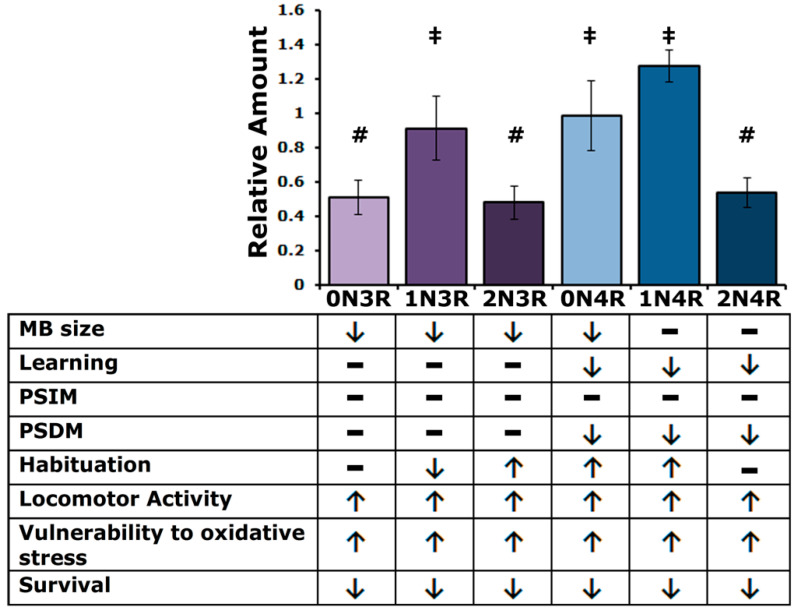
A summary of hTau steady state levels and isoform specific phenotypes as detailed in the text. Bars marked with # are not significantly different from each other but are from those marked with ‡ and conversely ‡ marked bars are not significantly different from each other but are from ones marked with #. Arrows denote changes in phenotype (up or down) compared to controls, while dashes denote no effect.

## Data Availability

Not applicable.

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
