# Peer review of "Differential Effects of Human Tau Isoforms to Neuronal Dysfunction and Toxicity in the Drosophila CNS"

_ijms, 2022, doi:10.3390/ijms232112985_

Round 1

Reviewer 1 Report

A straightforward and interesting study that provides needed basic information of the isoform specific actions of human tau on certain neurological functional characteristics of adult Drosophila. Experiments are well designed and figures are of high quality.

The manuscript is well written -- I found one typo: line 164 needs a reference?

Author Response

we thank the reviewer for being positive.

We have corrected the typo, thank you, and added the reference

Reviewer 2 Report

Authors presented the effects of six Tau isoforms on associative memory and learning, circadian activity, footshock habituation, development of mushroom body neurons, vulnerability to oxidative stress and life span.

All studies were conducted on the same model object, this is the advantage of the work.

Very appropriate, the authors provide statistical details in supplementary materials

But I have some remarks:

      1. The last sentence of the abstract is better to rephrase, it is not clear. (line 37-38)

      2. Introduction line 49-59: "Alternative splicing of the primary transcript from the single human MAPT gene on chromosome 17 gives rise to six isoforms (What? Tau?)  in the adult CNS ."

     3. In the Discussion after the sentence  «We have revealed a number of common and hTau isoform-specific effects in the Drosophila model…» (line 588-589)  it would be nice to list the identified patterns point by point.

Author Response

We thank the reviewer for the constructive comments and suggestions. Below are our responses.

  1. The last sentence of the abstract is better to rephrase, it is not clear. (line 37-38)

Done. We hope it is clear now

  1. Introduction line 49-59: "Alternative splicing of the primary transcript from the single human MAPT gene on chromosome 17 gives rise to six isoforms (What? Tau?) in the adult CNS …."

changed for clarity. We believe it is clear now

  1. In the Discussion after the sentence «We have revealed a number of common and hTau isoform-specific effects in the Drosophila model…» (line 588-589)  it would be nice to list the identified patterns point by point.

We believe we do, but they are all summarized in the last figure (Figure 9) as well.